# Application value of systemic immune-inflammation index in predicting severe *Mycoplasma pneumoniae* pneumonia

Xiaocong Guo[ID][1,2,3], Honglin Luo[3], Yaohui Song[3], Li Xiao[3], Xiaoya Xu[ID][4*], Yongcan Guo[1,2*]

**1** Department of Clinical Laboratory, The Affiliated Traditional Chinese Medicine Hospital, Southwest Medical University, Luzhou, China, **2** Department of Laboratory Medicine, The Affiliated Hospital of Southwest Medical University, Luzhou, China, **3** Department of Clinical Laboratory, Zigong First People's Hospital, Zigong, China, **4** Department of Neurology, Zigong First People's Hospital, Zigong, China

* xxy820122@163.com (XX); guoyongcan@swmu.edu (YG)

## Abstract

*Mycoplasma pneumoniae* (MP) is the primary causative agent of community-acquired pneumonia. Severe *mycoplasma pneumoniae* pneumonia (SMPP) can result in multiorgan damage and even respiratory failure or death. This study aimed to evaluate the predictive value of the Systemic Immune-Inflammation Index (SII) for SMPP. This retrospective study included 254 hospitalized children with MP infections (SMPP group, n = 103; non-SMPP group, n = 151). Patient data, including complete blood count parameters (white blood cell, absolute neutrophil, absolute lymphocyte, absolute monocyte, and platelet counts), C-reactive protein (CRP), serum amyloid A (SAA), and other markers, were collected. Furthermore, the SII, Systemic Inflammation Response Index (SIRI), neutrophil/lymphocyte ratio (NLR), monocyte/lymphocyte ratio (MLR), and platelet/lymphocyte ratio (PLR) were calculated. *T*-tests and the Mann–Whitney U test were used to analyze differences between the groups. Logistic regression was applied to analyze the risk factors. Receiver operating characteristic (ROC) curves were plotted to evaluate the predictive performance of the SII and CRP for SMPP. The SMPP group exhibited significantly higher CRP and SAA levels, SII, NLR, MLR, PLR, and SIRI than the non-SMPP group (all $P < 0.001$). Logistic regression revealed that the SII (odds ratio [OR] = 1.006, 95% confidence interval [CI]: 1.001–1.010) and CRP (OR = 1.080, 95% CI: 1.041–1.120) were independent risk factors. ROC curve of the SII (area under the ROC curve = 0.883, sensitivity = 0.699, and specificity = 0.881) outperformed that of CRP. Thus, SII can serve as an effective biomarker for SMPP prediction. It can be a rapid and cost-effective method when combined with routine blood tests, thereby demonstrating considerable potential for clinical application.

**Data availability statement:** The original contributions presented in the study are included in the article. All underlying data are freely available through supplementary files attached to this submission, ensuring full public access.

**Funding:** This work was supported in part by ZiGong Science and Technology Bureau, No.2023YKYXT06.

**Competing interests:** The authors declare that there are no conflict of interests.

## Introduction

Acute respiratory infection (ARI) is one of the most common human diseases. A 2002 study conducted by Williams et al. [1] reported that 1.6–2.2 million children succumb to ARI each year globally (95% confidence interval [CI]), of whom approximately 40% are in Africa and 30% in Asia. A 2014 epidemiological survey of pediatric ARIs in Guangzhou conducted by Liu Wenkuan et al. [2] indicated that Mycoplasma pneumoniae (MP) infection accounted for over 10% of all cases [3,4]. Since mid-2023, China has reported an atypical surge in pediatric MP infections, with severity rates escalating notably by September 2023 [5–7]. By 2024, the outbreak intensity necessitated public health interventions such as school closures in high-burden areas [7]. MP transmission occurs mainly through respiratory droplets in crowded settings with poor ventilation [7]. Notably, children aged <3 years were highly susceptible during the COVID-19 pandemic [8]. Compared with common MP infection, SMPP often progresses beyond pulmonary involvement, triggering extrapulmonary complications such as obstructive airway lesions, thromboembolic events, and multisystem inflammatory syndromes that substantially increase mortality risk. Furthermore, dysregulated host immunity drives SMPP pathogenesis, characterized by cytokine hyperproduction and collateral tissue injury; this can result in diseases that affect multiple body systems, such as the nervous, cardiovascular, and blood systems [9]. Once extrapulmonary complications occur, the disease can rapidly progress and develop into cytokine storm and respiratory distress syndrome, etc., potentially leading to respiratory failure and even death. Therefore, early identification of the severe progression trend of MP pneumonia (MPP) facilitates flexible adjustment of clinical treatment measures, and judgment of the disease process of MPP in children and control treatment nodes is of great importance [10].

A multicenter study conducted by Zhuo et al. [11] reported that the creatine kinase isoenzyme, lactate dehydrogenase (LDH), fibrinogen, and D-dimer levels, degree of fever as well as neutrophil/lymphocyte ratio (NLR) and platelet/lymphocyte ratio (PLR) were higher in the SMPP than in the non-SMPP group. In a comparative study between children with common MPP and those with SMPP, Yang et al. [12] found that the leukocyte and neutrophil counts as well as the levels of C-reactive protein (CRP), procalcitonin, interferon-γ, interleukin (IL)-2, IL-5, IL-6, IL-8, and IL-10 were significantly higher in the SMPP group. However, these indicators are not easy to obtain in children, and some take too long to detect. Shao et al. [13] analyzed the clinical value of novel inflammatory biomarkers in predicting MP infection and reported that the Systemic Immune-Inflammation Index (SII) and Systemic Inflammation Response Index (SIRI), which are based on routine blood results, can be used as supplementary diagnostic methods for MP infection. However, they did not study the predictive value of these biomarkers in clinical typing following infection.

SII is a comprehensive indicator based on three inflammatory cell types in the blood, namely, lymphocytes, neutrophils, and platelets. It is used to evaluate the immune and inflammatory status of the body using the formula SII = (neutrophil count × platelet count)/lymphocyte count [14]. The SII was originally proposed by Hu

et al. in 2014 to predict the prognosis of patients with hepatocellular carcinoma; since then, it has been widely used in the prognostic assessment of various tumors (e.g., small cell lung, ovarian, esophageal, and colorectal cancers) and other diseases [15–17]. The SII is considered to be more comprehensive than CRP in predicting the risk for coronary heart disease as it reflects both inflammation and thrombosis (platelet involvement) [18].

The SIRI is calculated as follows: SIRI = (neutrophil count × monocyte count)/lymphocyte count [14]. SIRI can comprehensively reflect the inflammatory state and immune balance of the body by integrating the absolute counts of neutrophils, monocytes, and lymphocytes. It was originally used to predict the prognosis of patients with malignant tumors (e.g., pancreatic cancer and nasopharyngeal carcinoma); however, its application has been extended to inflammatory diseases (e.g., severe pancreatitis and acute kidney injury) and cardiovascular diseases [19–21]. For example, in patients with gastric cancer, dynamic SIRI monitoring can predict the risk for recurrence, and patients demonstrating an increase in SIRI of more than 50% have significantly reduced survival [22]. In coronary heart disease, SIRI is associated with atherosclerotic plaque stability, and its elevation indicates active inflammation [19].

According to the evidence-based guideline for the diagnosis and treatment of Mycoplasma pneumoniae pneumonia in children (2023) [23], it can be diagnosed as SMPP if it meets any of the following manifestations, (1) Persistently high fever (> 39°C) ≥ 5d or fever ≥ 7d, with no downward trend in the peak temperature noted. (2) Wheezing, shortness of breath, dyspnea, chest pain, hemoptysis and so on. These manifestations are related to disease severity of the disease, combined with plastic bronchitis, asthma attack, pleural effusion, and pulmonary embolism (3) Extrapulmonary complications that do not meet the criteria for critical illness. (4) Oxygen saturation of finger pulse of≤ 0.93 when inhaling air at rest. (5) A single affected lung lobe involving 2/3 of the lobe. Moreover, uniform high-density consolidation is evident,or two or more lung lobes appear with high-density consolidation (regardless of the size of the affected area), which may be accompanied by a large amount of pleural effusion, and localized bronchiolitis performance. Diffuse or bilateral involvement of one lung with ≥ 4/ 5 lobar bronchiolitis coexisting with bronchitis, and mucus plug formation leading to atelectasis. (6) Progressively aggravated clinical symptoms were progressively aggravated, and the imaging revealing that the lesion range has progressed > 50% in 24–48 h. (7) One of CRP, LDH, or D-dimeris significantly increased. To further explore the practical prediction methods for severe cases, this study enrolled children with positive MP nucleic acid test results during the second half of 2024, who were then categorized into the SMPP (severe Mycoplasma pneumoniae pneumonia) group and non-SMPP group based on the aforementioned diagnostic criteria.

## Materials and methods

### Participants

This retrospective study included 254 hospitalized children with MPP (SMPP group, n = 103; non-SMPP group, n = 151) who were admitted to the Pediatrics Department of the First People's Hospital of Zigong City from July 1, 2024 to December 30, 2024. The classification into the SMPP and non-SMPP groups was made according to evidence-based guideline for the diagnosis and treatment of Mycoplasma pneumoniae pneumonia in children (2023) [23] Research data were accessed from the electronic medical record system on January 15, 2025.The exclusion criteria were as follows: (1) patients with comorbidities potentially confounding immune responses (e.g., genetic disorders, immunodeficiencies, or active malignancies); (2) patients with a previous history of neonatal respiratory distress syndrome, bacterial encephalitis, or other serious infectious diseases; and (3) patients with incomplete clinical data. The sample grouping and research protocol are illustrated in Fig 1.

A total of 628 children hospitalized with MPP between July 1 and December 31, 2024, were initially assessed for eligibility. After the application of exclusion criteria (n = 374), 254 patients were finally enrolled in this retrospective study. According to the 2023 evidence-based guideline for pediatric MPP, enrolled patients were classified into either the severe MPP (SMPP; n = 103) group or the non-severe MPP (non-SMPP; n = 151) group. The subsequent research protocol comprised data collection (clinical and laboratory parameters), computation of inflammatory indices—including the systemic

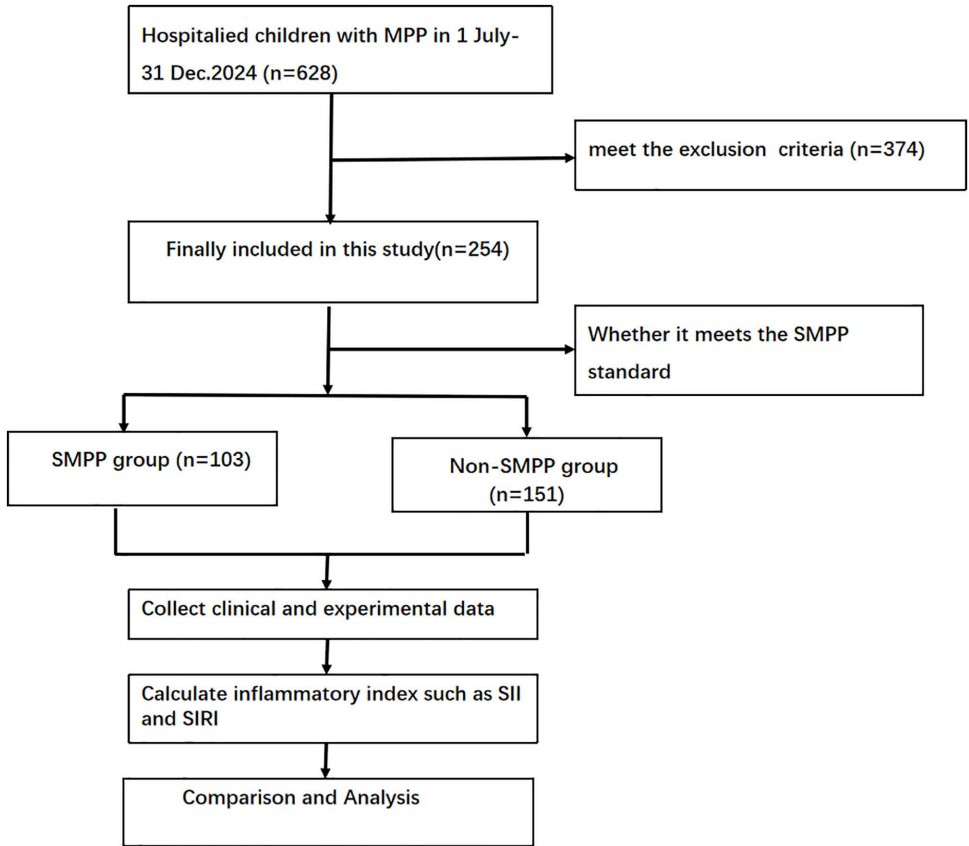

**Fig 1. Flowchart of patient recruitment, classification, and analysis for children with Mycoplasma pneumoniae pneumonia (MPP).**

immune-inflammation index (SII) and systemic inflammation response index (SIRI)—and comparative statistical analysis between groups.

Ethical oversight was provided by the Institutional Review Board (IRB) of Zigong First People's Hospital (Approval No, 2024−039). The IRB classified this study as exempt owing to its minimal risk nature and its exclusive dependence on the analysis of pre-existing, de-identified medical records. No new patient interviews were conducted as part of this specific data extraction and analysis study.

## Data collection

### General data

Patient data, including gender, age, length of hospital stay, maximum temperature, days of fever, chest X-ray plain film or chest computed tomography plain scan, and medical history, were collected. Furthermore, white blood cell (WBC) count, neutrophil count, lymphocyte count, monocyte count, platelet count, CRP, LDH, serum amyloid A (SAA), and other laboratory indicators were measured. The NLR, PLR, MLR, SII, and SIRI were also calculated.

### Statistical methods

Statistical analysis was conducted using the IBM SPSS version 26.0 (IBM Corp,Armonk,NY,USA) and GraphPad Prism 10.3(GraphPad software USA) software. The Shapiro-Wilk test was used to test the normality of the continuous variables.

Continuous variables with normal distribution were expressed as mean ± standard deviation and analyzed using the independent samples *t*-test. In contrast, continuous variables with non-normal distribution were expressed as median and interquartile range and analyzed using the Mann–Whitney U test. Categorical variables were expressed as frequencies (%) and analyzed using chi-squared tests. We first utilized univariate logistic regression analysis to analyze the factors affecting the severity of MPP in children. Then, for the statistically significant factors from the univariate regression results, multivariate logistic regression analysis was carried out to determine the independent risk factors for MPP severity in children. Receiver operating characteristic (ROC) curves were used to analyze the predictive power of the variables for SMPP. *P* < 0.05 was considered to indicate statistical significance.

## Results

### Comparison of general data and laboratory test results between the SMPP and non-SMPP groups

This study included 254 children with MPP (SMPP group, n = 103; non-SMPP group, n = 151). No significant difference was observed in sex, age, length of hospital stay, and Mono between the groups. The WBC count, neutrophil count, lymphocyte count, monocyte count, platelet count, CRP, SAA, SII, NLR, MLR, PLR, and SIRI were higher in the SMPP than in the non-SMPP group, with the difference being statistically significant (Table 1).

### Analysis of the risk factors

Univariate logistic regression analysis was conducted to identify the risk factors for SMPP (Table 2). The OR values of the SII, NLR, MLR, PLR, SIRI, CRP, and SAA were all greater than 1 (*P* < 0.001), indicating that the risk of increase in the aforementioned test indicators in patients with SMPP was significantly higher than that in non-SMPP patients. Multivariate logistic regression analysis revealed that after adjustment for other variables, the risks of elevated SII and CRP were significantly higher in patients with SMPP than in non-SMPP patients (odds ratio [OR] = 1.006, 95% confidence interval [CI]: 1.001–1.010, *P* = 0.008, and OR = 1.080, 95% CI: 1.041–1.120, *P* < 0.001, respectively). After adjustment for other

**Table 1. General information and inspection results.**

| Considerations | Nonserious (n = 151) | Severe (n = 103) | Z/χ² | P |
|---|---|---|---|---|
| Male | 74 (49%) | 47 (46.5) | 0.28 | 0.597 |
| Age | 5 (2,7) | 4 (3,7) | −0.038 | 0.97 |
| Days of hospitalization | 6 (4,7) | 6 (5,8) | −1.689 | 0.091 |
| Leukocyte (×10⁹/L) | 7.70 (5.69,10.07) | 9.17 (6.99, 13.21) | −4.092 | <0.001 |
| Neutrophil (×10⁹/L) | 3.66 (2.69, 4.79) | 5.83 (4.54,8.44) | −8.484 | <0.001 |
| Monocyte (×10⁹/L) | 0.67 (0.47,0.86) | 0.79 (0.53,1.12) | −2.450 | 0.014 |
| Lymphocyte (×10⁹/L) | 2.78 (1.83,4.36) | 1.95 (1.31, 3.23) | −3.729 | <0.001 |
| Platelet (×10⁹/L) | 280 (227, 335) | 327 (259, 436) | −3.201 | 0.001 |
| CRP (mg/L) | 5.30 (1.21, 11.6) | 24.83 (8.79, 43.6) | −8.123 | <0.001 |
| SAA (mg/L) | 35.26 (9.38, 110.12) | 135.41 (53.13, 217.71) | −6.928 | <0.001 |
| SII | 384.67(242.76, 569.28) | 899.22 (586.98, 1360.86) | −10.361 | <0.001 |
| NLR | 1.33(0.83,2.03) | 3.13 (1.94,4.03) | −9.263 | <0.001 |
| MLR | 0.23 (0.17,0.32) | 0.36(0.26,0.50) | −6.216 | <0.001 |
| PLR | 94.92 (73.53, 141.03) | 144.12 (112.23, 211.7) | 6.654 | <0.001 |
| SIRI | 0.79 (0.48,1.50) | 2.05 (1.47,3.72) | −9.473 | <0.001 |

Note: *P* < 0.05 difference indicates statistical significance. Data presented as median (interquartile range) for continuous variables and frequency (%) for categorical variables. Abbreviations: CRP, C-reactive protein; SAA, serum amyloid A; SII, systemic immune inflammatory index; NLR, neutrophil/lymphocyte ratio; MLR, monocyte/lymphocyte ratio; PLR, platelet/lymphocyte ratio; SIRI, systemic inflammatory response index.

factors, the SII and CRP remained as independent risk factors for SMPP, which can accurately predict the risk of SMPP. In contrast, the NLR, MLR, PLR, SIRI, and SAA could not effectively predict the risk of SMPP ($P > 0.05$), demonstrating no statistical significance.

### ROC curve analysis

ROC curves were plotted for the independent risk factors SII and CRP obtained from the multivariate regression analysis in Table 2 (Fig 2) to compare the area under the ROC curve (AUC), sensitivity, and specificity of the two factors (Table 3).

**Table 2.  Logistic model of SMPP occurrence.**

| Considerations | One-way regression analysis | | | | Multivariate regression analysis | | | | Cohen's d |
|---|---|---|---|---|---|---|---|---|---|
| | β | SE | OR (95% CI) | *P* | β | SE | OR (95% CI) | *P* | |
| SII | 0.005 | 0.001 | 1.005 (1.004,1.007) | <0.001* | 0.006 | 0.002 | 1.006 (1.001,1.010) | 0.008* | 1.12 |
| NLR | 1.244 | 0.167 | 3.471 (2.503, 4.813) | <0.001* | 0.176 | 0.337 | 0.838 (0.433,1.623) | 0.601 | 1.19 |
| MLR | 5.059 | 0.939 | 157.505 (25.028,991.214) | <0.001* | 0.309 | 3.792 | 0.734 (0,1240.518) | 0.734 | 0.83 |
| PLR | 0.014 | 0.002 | 1.014 (1.009,1.018) | <0.001* | 0.001 | 0.008 | 1.001 (0.985,1.017) | 0.908 | 0.82 |
| SIRI | 1.305 | 0.189 | 3.689 (2.549,5.339) | <0.001* | 0.200 | 0.694 | 1.221 (0.313, 4.762) | 0.774 | 0.93 |
| CRP | 0.078 | 0.012 | 1.081 (1.055,1.107) | <0.001* | 0.077 | 0.019 | 1.080 (1.041,1.120) | <0.001* | 1.1 |
| SAA | 0.011 | 0.002 | 1.011 (1.007,1.014) | <0.001* | 0.002 | 0.003 | 1.002 (0.996,1.009) | 0.48 | 0.96 |

Note: *$P < 0.05$ indicates statistical significance. Abbreviations: CRP, C-reactive protein; SAA, serum amyloid A; SII, systemic immunoinflammatory index; NLR, neutrophil/lymphocyte ratio; MLR, monocyte/lymphocyte ratio; PLR, platelet/lymphocyte ratio; SIRI, systemic inflammatory response index.

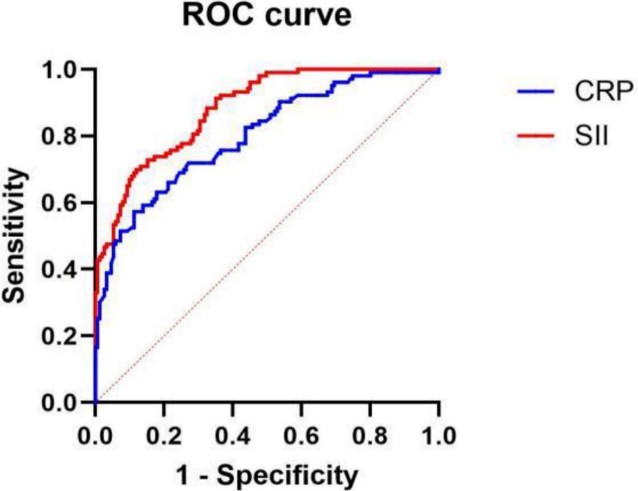

**Fig 2.  Receiver operating characteristic (ROC) curves for the Systemic Immune-Inflammation Index (SII) and C-reactive protein (CRP) in predicting severe Mycoplasma pneumoniae pneumonia (SMPP) in children.**

**Table 3. Predicting the performance of SII and CRP on SMPP.**

| Considerations | AUC | Best cutoff point | Sensitivity | Specificity |
|---|---|---|---|---|
| SII | 0.883(0.844,0.922) | 689.85 | 0.699 | 0.881 |
| CRP | 0.800(0.745,0.855) | 19.2 | 0.573 | 0.887 |

Note: AUC, area under the ROC curve.

ROC curves were generated based on the independent risk factors identified by multivariate logistic regression analysis (see Table 2 for regression results). The performance metrics, including area under the curve (AUC) with 95% confidence intervals, sensitivity, specificity, and optimal cutoff values for both SII and CRP are presented in Table 3.

The results indicated that for the SII, the AUC was 0.883 (95% CI: 0.884–0.922); sensitivity, 0.699; specificity, 0.881; and optimal cutoff value, 689.85. Meanwhile, for CRP, the AUC was 0.800 (95% CI: 0.745–0.855); sensitivity, 0.573; specificity, 0.887; and optimal cutoff value, 19.2. Therefore, the SII is better than CRP in predicting SMPP.

## Discussion

In 2023, the Respiratory Group of the Pediatrics Society of the Chinese Medical Association issued the Evidence-based Guidelines for Diagnosis and Treatment of MPP in Children. The diagnosis of SMPP is mainly based on persistent fever, severe clinical symptoms, imaging findings, and laboratory indicators. After contracting MP infection, pneumonia symptoms usually appear 2–3 days after fever [23]. Children with MP infection typically present with persistent cough and fever (often ≥39°C);however, afebrile or low-grade fever cases occur [7,23]. A distinguishing feature is the progression from the initial paroxysmal dry cough to productive sputum [7]. This suggests that most clinical symptoms are nonspecific and that progressive worsening of clinical symptoms requires an experienced pediatrician and differentiation from other respiratory disorders. Compared with the indicators of laboratory inflammation, it can more intuitively reflect the degree of inflammation in the human body. The guidelines suggest that if any one of the three indicators, namely CRP, LDH, and D-dimer, shows a significant elevation, there is a risk of developing severe mycoplasma pneumonia (SMPP). However, these indicators are not as widely available and accessible to children as blood tests. Blood tests are rapid and inexpensive. They play a pivotal role in the diagnosis of almost all diseases. In recent years, novel inflammatory biomarkers, such as NLR, MLR, PLR, SII, and SIRI, which are calculated based on blood routine analysis results, have become research hotspots [13,24,25]. These inflammatory biomarkers, calculated using the ratio of different components, provide a more comprehensive and accurate picture of inflammatory status than the single parameter of complete blood count.

Previous studies have indicated that SMPP pathogenesis involves MP-induced direct tissue damage and dysregulated host immunity, leading to cytokine hyperproduction and neutrophil-mediated inflammation [25–27]. SII integrates platelet, neutrophil, and lymphocyte counts to reflect systemic inflammation. Neutrophils play a key role in the progression of pneumonia [28]. MPP membrane lipoproteins bind alveolar macrophage Toll-like receptors, activating NF-κB signaling [28,29]. This induces proinflammatory cytokines (e.g., IL-8, TNF-α, GM-CSF), driving neutrophil recruitment and phagocytosis [29]. Ultimately, excessive inflammation impairs immune function [29–31]. Relevant studies have demonstrated that PLT is positively correlated with the length of hospital stay, mortality, and disease prognosis in patients with MPP [32,33]. In a retrospective study of various inflammatory indicators involving 304 children with MPP, Wang et al. [25] found that the SII can be used as a predictor of disease severity in children with MPP and that its predictive effect is better than that of NLR, SIRI, and PLR. Our study systematically evaluated the application value of inflammatory markers in SMPP prediction. The results indicated that the SII had an excellent predictive power for SMPP. The SII was significantly higher in the SMPP than in the non-SMPP group ($P<0.001$). Multivariate regression confirmed that the SII was an independent risk factor (OR = 1.006, 95% CI: 1.001–1.010). The AUC (0.883) was better than traditional inflammatory markers, such as CRP, and other inflammatory markers, such as SIRI, NLR, and PLR.

The predictive advantage of SII is due to its comprehensive quantification of the synergistic effects of platelets, neutrophils and lymphocytes [34–37]. This study determined for the first time that the optimal critical value of SII to predict the severity of SMPP was 689.85 (sensitivity, 0.699; specificity, 0.881), and its abnormal increase suggested excessive immune activation, which was consistent with the cytokine storm mechanism of SMPP. This critical value can effectively identify the risk of severe cases and reduce misdiagnosis.

As a traditional inflammatory marker, although CRP is an independent risk factor (OR = 1.080), its AUC is only 0.800, and its sensitivity is relatively low (0.573). This may be because CRP reflects the late inflammatory response. Compared with CRP, as a systemic immune inflammatory indicator, SII is better able to capture the early signals of immune imbalance. Although SAA was significantly higher in the SMPP group, multivariate regression revealed that its predictive value was limited, indicating that it is difficult to comprehensively evaluate the risk of severe disease using a single index. In clinical practice, the SII calculation is simple and based on blood routine examination data, which is economical and timely. If the SII is included in the early evaluation system of MP-infected children, it may assist clinicians in identifying groups at a high risk for SMPP in a timely and accurate manner, optimizing treatment strategies, and reducing the incidence of severe diseases.

This study was a single-center retrospective analysis and may demonstrate selection bias. In the future, multicenter prospective studies should be conducted to further verify the SII in early warning, efficacy monitoring, and prognosis evaluation of SMPP in combination with molecular biological detection.

## Conclusion

The results reveal that SII can serve as an effective biomarker for predicting SMPP, demonstrating superior predictive performance compared with traditional inflammatory markers. Considering the rapid and cost-effective nature of routine blood tests, SII holds significant clinical value as an early warning of SMPP.

## Supporting information

**S1 File. Raw data for evaluating the value of SII in predicting severe mycoplasma pneumoniae pneumonia in children.**
(XLSX)

## Acknowledgments

The authors gratefully acknowledge the intense individual effort and support from many sources to make this study possible.

## Author contributions

**Conceptualization:** Xiaocong Guo, Xiaoya Xu.

**Data curation:** Xiaocong Guo, Honglin Luo, Li Xiao.

**Formal analysis:** Xiaocong Guo, Honglin Luo.

**Funding acquisition:** Xiaocong Guo.

**Investigation:** Xiaocong Guo, Yaohui Song.

**Methodology:** Xiaocong Guo, Yaohui Song, Honglin Luo.

**Project administration:** Xiaoya Xu.

**Resources:** Xiaocong Guo, Li Xiao.

**Software:** Honglin Luo.

**Supervision:** Yongcan Guo.

**Validation:** Yaohui Song.

**Visualization:** Xiaoya Xu.

**Writing – original draft:** Xiaocong Guo.

**Writing – review & editing:** Yongcan Guo.

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
