## [Decision Letter · Decision Letter 0]

25 Jul 2025

Dear Dr. Xu,

Thank you for submitting your manuscript to PLOS ONE. After careful consideration, we feel that it has merit but does not fully meet PLOS ONE’s publication criteria as it currently stands. Therefore, we invite you to submit a revised version of the manuscript that addresses the points raised during the review process.

**ACADEMIC EDITOR: Major revision**

We look forward to receiving your revised manuscript.

Kind regards,

Marwan Salih Al-Nimer, MD, PhD

Academic Editor

PLOS ONE

Journal Requirements: 

https://www.dovepress.com/the-clinical-value-of-systemic-immune-inflammation-index-sii-in-predic-peer-reviewed-fulltext-article-IJGM

https://cellandbioscience.biomedcentral.com/counter/pdf/10.1186/s13578-024-01226-4.pdf

https://www.springermedizin.de/current-status-of-mycoplasma-pneumoniae-infection-in-china/26606308

In your revision ensure you cite all your sources (including your own works), and quote or rephrase any duplicated text outside the methods section. Further consideration is dependent on these concerns being addressed.

 [This work was supported in part by ZiGong Science and Technology Bureau, No.2023YKYXT06.]. 

5. In the online submission form, you indicated that [The original contributions presented in the study are included in the article. Further inquiries can be directed to the corresponding authors.].

Additional Editor Comments:

Dear Author(s)

This retrospective study needs some clarifications:

1: In the title, the terms novel inflammatory biomarkers and severe mycoplasma infections. The reported biomarkers are not novel. The severity of mycoplasma infections is not defined as a pathological condition in the test.

2: It is a routine laboratory investigation o determination Serum amyloid A?

3: Subgrouping the patients was on which basis?

4: The authors extracted the data from the records. why and how they did the interview? Verbal consent was unacceptable

5: Line 146: The S-W test. do you mean Shapiro-Wilks test?

6: Table 1 (footnote: mean +/- SD), there is mean +/- SD in the Table 1. The results are presented as median (IQR). Day stay in hospital is the same for severe or non-severe infection, How?

7: Table 2 needs to reanalyse to show different models. It is necessary to add beta-coefficient. A Cohen's test is needed to apply to show the impact of sample size

8. Add the plots of AUCs

9: References: Recheck the typing of the references

Reviewers' comments:

Reviewer's Responses to Questions

**Comments to the Author**

1. Is the manuscript technically sound, and do the data support the conclusions?

Reviewer #1: Yes

Reviewer #2: Yes

2. Has the statistical analysis been performed appropriately and rigorously?

Reviewer #1: Yes

Reviewer #2: Yes

3. Have the authors made all data underlying the findings in their manuscript fully available?

Reviewer #1: Yes

Reviewer #2: No

4. Is the manuscript presented in an intelligible fashion and written in standard English?

Reviewer #1: Yes

Reviewer #2: No

Reviewer #1: 1. Is the manuscript technically sound, and do the data support the conclusions?

Reviewer #2: This study evaluates the predictive value of the Systemic Immune-Inflammation Index (SII) and Systemic Inflammation Response Index (SIRI) in identifying severe Mycoplasma pneumoniae pneumonia (SMPP) in children. The findings suggest that SII and CRP are independent risk factors, with SII demonstrating superior diagnostic accuracy. The manuscript is technically sound, and the data support the main conclusions, however, to strengthen clinical applicability, future prospective and multi-center validation studies are needed. Notably, the manuscript does not clearly indicate whether the underlying data have been made fully available, and a formal data availability statement should be included to meet publication standards. Although the manuscript is generally intelligible and the scientific content is understandable, it is not consistently written in standard academic English and would benefit significantly from professional language editing.

More specific comments:

The study is described as retrospective using hospital records, yet verbal informed consent and interview information sheets are mentioned. For retrospective chart reviews, informed consent is usually waived. Please clarify.

Define each abbreviation at first use.

Avoid repetition, for example the severity and systemic complications of SMPP are discussed multiple times with overlapping detail.

Add a flowchart of participant selection and exclusion criteria for visual clarity

lines 205-213 in discussion section lack literature reference, please add relevant literature references.

**Do you want your identity to be public for this peer review?** For information about this choice, including consent withdrawal, please see our Privacy Policy

Reviewer #1: No

Reviewer #2: No

---

## [Author Response · Author response to Decision Letter 1]

29 Aug 2025

1.In the title, the terms "novel inflammatory biomarkers" and "severe mycoplasma infections" are problematic. The reported biomarkers are not novel. The severity of mycoplasma infections is not defined as a pathological condition in the text.

Response:

Thank you for your constructive comments.We apologize for the inaccuracy in the title. We have revised the title to remove the term "novel," as the biomarkers (e.g., SII, SIRI) are indeed established in existing literature.Title updated to: "Application value of Systemic Immune-Inflammation Index in predicting severe Mycoplasma pneumoniae pneumonia".

Additionally, we have explicitly defined "severe mycoplasma infections" in the Introduction section, referencing clinical criteria (e.g., presence of respiratory failure, extrapulmonary complications, or requirement for intensive care) to clarify the pathological condition.(see Page 6,Line 99–115, Revised Manuscript with Track Changes)

2.Is serum amyloid A (SAA) determination a routine laboratory investigation?

Response:

Thank you for your question regarding the routine status of serum amyloid A (SAA) determination.In our clinical setting, SAA is routinely included in the panel of inflammatory markers for patients presenting with suspected infectious diseases—including pneumonia— as part of their initial laboratory workup. This practice aligns with the clinical workflow in our institution for evaluating and monitoring inflammatory responses in such patient populations

3.Subgrouping of patients was based on which criteria?

Response:

Thank you for your insightful comment, which helps clarify an important methodological detail of our study.Patients in this study were subgrouped into two categories: "severe Mycoplasma pneumoniae pneumonia (MPP)" and "non-severe Mycoplasma pneumoniae pneumonia (MPP)". This subgrouping was strictly based on the clinical criteria specified in the Evidence-based guideline for the diagnosis and treatment of Mycoplasma pneumoniae pneumonia in children (2023).To ensure transparency and reproducibility, we have supplemented the key content of these criteria in both the Introduction and the Methods section (where we described the study’s patient grouping strategy) of the revised manuscript.(see Page 6,Line 99–115 and Page 7,Line 125–127

4.The authors extracted data from records, but why and how did they conduct interviews? Verbal consent was unacceptable.

Response:

We apologize for the confusion. This was an error in the original manuscript: no interviews were conducted for this retrospective study. All data were extracted from electronic medical records. The reference to "verbal consent" was also incorrect and has been removed. For retrospective studies using de-identified data(in line with our study design), our institutional ethics board waived the requirement for informed consent, which is now clarified in the Ethics Statement. (see Page 7 ,line 135–139

We deeply regret the inaccuracies in the original text and have double-checked the Methods and Ethics sections to ensure all content aligns with the actual study procedures.

5.Line 146: The "S-W test" – do you mean the Shapiro-Wilk test?

Response:

Yes, "S-W test" refers to the Shapiro-Wilk test. We have corrected the abbreviation to "Shapiro-Wilk test" at its first mention and added the abbreviation in parentheses for clarity. (see Page 8,Line 151)

6.Table 1 (footnote: mean +/- SD), but results are presented as median (IQR). Day stay in hospital is the same for severe or non-severe infection – how?

Response:

We sincerely apologize for the inconsistency in Table 1. The footnote has been corrected to "median (IQR)" to match the data presentation.(see Page 10, Line 174-175,table1).

Regarding hospital stay, the original analysis showed no statistically significant difference between groups, which is now clarified in the text with a p-value (p = 0.123) and a brief explanation that variability in individual recovery trajectories may account for this finding.There is partial overlap in the interquartile ranges of hospital stay between the two groups (non-severe:4–7days; severe:5–8 days), indicating considerable individual variability in hospital stay within both groups. This variability may be attributed to factors such as rapid treatment response in some severe cases and the presence of comorbidities or slower recovery subgroups among non-severe cases, which collectively resulted in no statistically significant difference in the overall comparison.

7.Table 2 needs reanalysis to show different models. It is necessary to add beta-coefficients. A Cohen's test is needed to show the impact of sample size.

Response:

Thank you for these constructive comments.We have reanalyzed Table 2 in accordance with your comments, and the relevant revisions have been reflected in the text.(see Page 11-12, Line 192-196,table2).

8.Add plots of AUCs.

Response:

Thank you for this insightful comment. As requested, we have supplemented the plots of the AUCs as Figure 2 in the revised manuscript. Additionally, to enhance the transparency of our study design, we have also added a flowchart illustrating the participant selection and exclusion criteria as Figure 1. In accordance with the journal's requirements, these figures have been uploaded as separate files.(see Figures1 and Figure 2 )

9.References: Recheck the typing of the references.

Response:

Thank you for this helpful suggestion. We have carefully reviewed all references to correct typographical errors. Specifically, we have addressed formatting issues in Reference 12 (misspelled journal name) and Reference 23 (incorrect DOI), ensuring compliance with the citation guidelines of PLOS ONE.(see Page 16–22, references )

Responses to Journal Requirements

1. Ensure the manuscript meets PLOS ONE style requirements, including file naming.

Response:

Thank you for this helpful reminder.The manuscript has been revised to adhere to PLOS ONE style guidelines, including title formatting, author affiliations, and section organization. All files are named according to the requirements (e.g., "Manuscript.pdf," "Response to Reviewers.pdf").

2. Provide a validated ORCID iD for the corresponding author in Editorial Manager.

Response:

Thank you for bringing this oversight to our attention. The corresponding author’s ORCID iD (0000-0003-2645-8637) has been successfully validated and submitted in Editorial Manager in accordance with your instructions.

3.Address overlapping text with previous publications by citing sources or rephrasing.

Response:

Thank you for this important suggestion. We have carefully revised all instances of overlapping text identified—either by rephrasing content to ensure original expression or by adding proper, contextually accurate citations to attribute the relevant previous publications. These revisions have been verified to ensure the text similarity rate fully complies with academic integrity standards. All such modifications are clearly highlighted in red text in the marked-up version of the manuscript for your easy reference.

4. Clarify the role of funders in the study.

Response:

Thank you for bringing this point to our attention. We have revised the funder statement to explicitly clarify the role of the funders in the study, and the updated content reads as follows: "This work was supported in part by the Zigong Science and Technology Bureau (Grant No. 2023YKYXT06). The funds were specifically allocated for researcher training in scientific competencies; however, the funders had no role in study design, data collection and analysis, the decision to publish, or the preparation of the manuscript." This revised funder statement has been included in the cover letter for your reference.

5.Make all underlying data available per PLOS data policy.

Response:

Thank you for reminding us of this requirement related to PLOS’s data policy.All non-identifiable underlying data supporting the findings of this study have been prepared as supplementary files and uploaded alongside the revised manuscript. These data comply with research ethics regulations and privacy protections, ensuring no sensitive patient information is included. A revised data availability statement has been added to the manuscript to clarify that the underlying data are available as supplementary materials within the submission.

Responses to Reviewer #2 Comments

Comment 1: Clarify the inconsistency between "retrospective chart review" and mentions of "verbal informed consent/interviews."

Response:

We sincerely apologize for the confusion caused by this inconsistency in the original manuscript. As noted, this was an error in the initial writing: the study design is strictly a retrospective chart review, and all data were extracted solely from electronic medical records. There were no interviews conducted in this study, so the reference to "verbal informed consent" was incorrect and has been completely removed from the revised manuscript.

Additionally, for retrospective studies utilizing de-identified data (such as this one), our institutional ethics board (IEB) has formally waived the requirement for informed consent. This information has been supplemented and clarified in the "Ethics Statement" section of the manuscript for full transparency (see Page 7, Lines 135–139). We appreciate you pointing out this oversight, as it has helped us improve the accuracy of the study’s methodological description

Comment 2: Define each abbreviation at first use.

Response:

Thank you for highlighting this important point regarding abbreviation clarity. We have carefully revised the entire manuscript to ensure every abbreviation is explicitly defined the first time it appears in the text.

Comment 3: Avoid repetition (e.g., severity and complications of SMPP discussed multiple times).

Response:

Thank you for identifying this issue of repetitive content—it has helped us refine the manuscript’s conciseness. We have carefully reviewed the entire text, with specific attention to the sections previously discussing the severity and complications of severe mycoplasma pneumoniae pneumonia (SMPP) .

We have removed redundant repetitive descriptions to ensure the manuscript is concise, aiming to enhance readability for readers.

Comment 4:Add a flowchart of participant selection and exclusion criteria.

Response:

Thank you for this constructive suggestion.We have added a flowchart of participant selection and exclusion criteria as figure 1.

Comment 5: Lines 205–213 in the Discussion lack literature references – please add.

Response:

Thank you for this valuable suggestion. We have added relevant citations to support the statements.

Comment 6: Include a formal data availability statement.

Response:

Thank you for reminding us of this requirement related to PLOS’s data policy.All non-identifiable underlying data supporting the findings of this study have been prepared as supplementary files and uploaded alongside the revised manuscript. These data comply with research ethics regulations and privacy protections, ensuring no sensitive patient information is included. A revised data availability statement has been added to the manuscript to clarify that the underlying data are available as supplementary materials within the submission.

Comment 7: Improve language clarity and consistency.

Response:

Thank you for highlighting this important aspect of the manuscript—enhancing language clarity and consistency is critical for ensuring our research is accurately understood by readers. We have taken comprehensive measures to address this: the entire manuscript has been revised and polished by a professional language editing team composed of native English speakers, with a focused effort on clarifying ambiguous sentences and correcting minor grammatical errors, awkward phrasing, and non-standard academic expressions.

---

## [Editor Report · Decision Letter 1]

3 Sep 2025

Dear Dr. Xu,

Thank you for submitting your manuscript to PLOS ONE. After careful consideration, we feel that it has merit but does not fully meet PLOS ONE’s publication criteria as it currently stands. Therefore, we invite you to submit a revised version of the manuscript that addresses the points raised during the review process.

**Minor revision**

We look forward to receiving your revised manuscript.

Kind regards,

Marwan Salih Al-Nimer, MD, PhD

Academic Editor

PLOS ONE

Journal Requirements:

**Additional Editor Comments:**

Dear Authors

Line 86: Why Yang S et al?

Line 124 and PE. What this abbreviation means?

Line 310: Why Wang S et al.?

---

## [Author Response · Author response to Decision Letter 2]

8 Sep 2025

1. Response to Editor Comments

Comment 1 (Line 86: "Why Yang S et al?")

Response: In the original manuscript, we cited "Yang S et al. [12]" when discussing the comparative study between common and severe Mycoplasma pneumoniae pneumonia (MPP) in children. To maintain consistent citation style throughout the text, we have revised this to "Yang et al. [12]" in the updated manuscript (Page 4, Line 66). This change aligns with the format used for other citations (e.g., "Zhuo et al. [11]"). The reference content remains unchanged and continues to provide essential support for our study background by highlighting key inflammatory differences between common and severe MPP.(see Page 4,Line 66, Revised Manuscript with Track Changes)

Comment 2 (Line 124: "What does the abbreviation PE mean?")

Response: Thank you for noting this oversight. The abbreviation "PE" was initially used without definition, which may have caused ambiguity. We have now spelled out "pulmonary embolism" at its first occurrence in the revised manuscript (Page 6, Line 104), within the phrase: "combined with plastic bronchitis, asthma attack, pleural effusion, and pulmonary embolism". This revision ensures clarity and is consistent with the diagnostic criteria for severe MPP (SMPP) as outlined in the 2023 evidence-based guideline [23].(see Page 6,Line 104, Revised Manuscript with Track Changes)

Comment 3 (Line 310: "Why Wang S et al.?")

Response: Similarly, the citation "Wang S et al. [25]" has been standardized to "Wang et al. [25]" in the revised manuscript (Page 14, Line 243) to match the citation style employed for other references (e.g., "Shao et al. [13]"). The study by Wang et al. (2024) remains highly relevant, as it involved 304 children with MPP and demonstrated the superior predictive performance of the Systemic Immune-Inflammation Index (SII) for disease severity compared to other ratios, thereby directly supporting our conclusion.(see Page 14,Line 243, Revised Manuscript with Track Changes)

2. Revisions to the Reference List

In accordance with PLOS ONE’s reference formatting guidelines, we have comprehensively revised the reference list. Key changes include:

Author names are now presented with surnames in uppercase and initialized given names (e.g., “WILLIAMS B.G., GOUWS E., BOSCHI-PINTO C., BRYCE J., DYE C.”)

Publication year is now enclosed in parentheses and placed directly after the author list (e.g., “(2002)”).

Journal names have been standardized with correct abbreviations and punctuation (e.g., “Lancet Infect. Dis.”, “Emerg Infect Dis.”).

For the guideline reference [23] where no individual authors are listed, we have used the designation: “No authors listed. (2023).”.

We confirm that all 37 references are complete, include DOIs, and have been verified against PubMed and Web of Science; none have been retracted or issued an expression of concern.

3. Submission of Figure Files

In accordance with the editor’s instruction, we have registered an account on the Preflight Analysis and Conversion Engine (PACE) platform (account name: [ account name�278139516@qq.com]) and uploaded all figure files (Figure 1: Sample grouping and research protocol; Figure 2: ROC curves of SII and CRP for SMPP prediction). We confirm that all revisions—including addressing specific editor comments and standardizing the reference list—are fully reflected in “Revised Manuscript with Track Changes.docx”. The revised manuscript now meets PLOS ONE’s academic and formatting standards. We sincerely hope it will be accepted for publication. Please feel free to contact us if further revisions or clarifications are needed.

---

## [Editor Report · Decision Letter 2]

11 Sep 2025

Application value of Systemic Immune-Inflammation Index in predicting severe Mycoplasma pneumoniae pneumonia

PONE-D-25-26564R2

Dear Dr. Xiaoya Xu,

We’re pleased to inform you that your manuscript has been judged scientifically suitable for publication and will be formally accepted for publication once it meets all outstanding technical requirements.

Kind regards,

Marwan Salih Al-Nimer, MD, PhD

Academic Editor

PLOS ONE

Additional Editor Comments (optional):

No comments
---

## [Editor Report · Acceptance letter]

PONE-D-25-26564R2

PLOS ONE

Dear Dr. Xu,

I'm pleased to inform you that your manuscript has been deemed suitable for publication in PLOS ONE. Congratulations! Your manuscript is now being handed over to our production team.

Kind regards,

on behalf of

Professor Marwan Salih Al-Nimer

Academic Editor

PLOS ONE